



# Global and regional impacts of land cover changes on isoprene emissions derived from spaceborne data and the MEGAN model

Beata Opacka[1], Jean-François Müller[1], Trissevgeni Stavrakou[1], Maite Bauwens[1], Katerina Sindelarova[2], Jana Markova[2,3], and Alex B. Guenther[4]

[1]Royal Belgian Institute for Space Aeronomy (BIRA-IASB), Avenue Circulaire 3, 1180 Brussels, Belgium
[2]Charles University in Prague, Department of Atmospheric Physics, Prague, Czech Republic
[3]Czech Hydrometeorological Institute (CHMI), Na Šabatce 17, 14306, Prague 4, Czech Republic
[4]Department of Earth System Science, University of California Irvine, 92697, California, USA

*Correspondence to*: Beata Opacka (beata.opacka@aeronomie.be)

**Abstract.** Among the biogenic volatile organic compounds (BVOCs) emitted by plant foliage, isoprene is by far the most important in terms of both global emission and atmospheric impact. It is highly reactive in the air, and its degradation favours the generation of ozone (in presence of $NO_x$) and secondary organic aerosols. A critical aspect of BVOC emission modelling is the representation of land use and land cover (LULC). The current emission inventories are usually based on land cover maps that are either modelled and dynamic or satellite-based and static. In this study, we use the state-of-the-art MEGAN model coupled with the canopy model MOHYCAN to generate and evaluate emission inventories relying on satellite-based LULC maps at annual time steps. To this purpose, we first intercompare the distribution and evolution (2001 – 2016) of tree coverage from three global satellite-based datasets, MODIS, ESA CCI-Land Cover (ESA CCI-LC) and the Global Forest Watch (GFW), and from national inventories. Substantial differences are found between the datasets, e.g. the global areal coverage of trees ranges from 30 to 50 Mkm², with trends spanning from -0.26% yr$^{-1}$ to +0.03% yr$^{-1}$ between 2001 and 2016. At national level, the increasing trends in forest cover reported by some national inventories (in particular for the US) are contradicted by all remotely-sensed datasets. Three inventories of isoprene emissions are generated, differing only in their LULC datasets used as input: (i) the static distribution of the stand-alone version of MEGAN, (ii) the time-dependent MODIS land cover dataset, and (iii) the MODIS dataset modified to match the tree cover distribution from the GFW database. The mean annual isoprene emissions (350 – 520 Tg yr$^{-1}$) span a wide range due to differences in tree distributions, especially in isoprene-rich regions. The impact of LULC changes is a mitigating effect ranging from 0.04 to 0.33% yr$^{-1}$ on the positive trends (0.94% yr$^{-1}$) mainly driven by temperature and solar radiation. This study highlights the uncertainty in spatial distributions and temporal variability of isoprene associated to remotely-sensed LULC datasets. The interannual variability of the emissions is evaluated against spaceborne observations of formaldehyde (HCHO), a major isoprene oxidation product, through simulations using the global chemistry-transport model (CTM) IMAGESv2. A high correlation (R>0.8) is found between the observed and simulated interannual variability of HCHO columns in most forested regions. The implementation of LULC change has little impact on this correlation, due to the dominance of meteorology as driver of short-term interannual variability. Nevertheless, the simulation accounting for the large tree cover declines of the GFW database over several regions,



notably Indonesia and Mato Grosso in Brazil, provides the best agreement with the HCHO column trends observed by OMI. Overall, our study indicates that the continuous tree cover fields at fine resolution provided by the GFW database are our

preferred choice for constraining LULC (in combination with discrete LULC maps such as those of MODIS) in biogenic isoprene emission models.

## 1 Introduction

The total biogenic volatile organic compounds (BVOCs) emission into the atmosphere amounts to ca. 1,000 Tg yr$^{-1}$ of which about 50-60% of the share is entailed by isoprene (Lathière et al., 2006; Guenther et al., 2012; Sindelarova et al., 2014; Messina

et al., 2016; Granier et al., 2019) and is roughly equal to the global methane emission (Lelieveld et al., 1998; Saunois et al., 2020). Isoprene is highly reactive and affects tropospheric chemistry (Fehsenfeld et al. 1992; Atkinson et al., 2000; Pike and Young, 2009). Under high-NO$_x$ conditions (NO$_x \equiv$ NO + NO$_2$), isoprene is a major precursor of tropospheric ozone (Atkinson et al., 2000; Ryerson et al., 2001; da Silva et al., 2018; Mo et al., 2018; Saunier et al., 2020). BVOCs also affect the growth of secondary organic aerosols (Claeys et al., 2004; Kroll et al., 2005 & 2006; Carlton et al., 2009) and influence tropospheric

hydroxyl radicals (OH) levels through depletion or regeneration (Lelieveld et al., 2008; Hofzumahaus et al., 2009; Fuchs et al., 2013; Hansen et al., 2017), thereby altering the lifetime of methane. Isoprene is primarily emitted from terrestrial vegetation, in particular broadleaf trees, and therefore, isoprene emissions are critically dependent on the land cover (e.g. tree, shrub, grass, crop) and on the plant species within those land covers (Arneth et al., 2011; Guenther et al., 2012).

Land use and land cover changes (LULCCs) are considered among the main drivers of environmental and climate changes (Foley et al., 2005; Turner et al., 2007; Jia et al., 2019). They bring about disruption in land-atmosphere interactions through multiple biophysical and biogeochemical fluxes across different spatial and temporal scales. In particular, the impact of deforestation on the climate system was reviewed by Bonan et al. (2008), Unger et al. (2014), Scott et al. (2018) and Zeppetello et al. (2020). Distribution and disturbances in vegetation, in particular trees, impact the emissions of BVOC that in turn control

the loadings of several short-lived climate forcers, with effects on climate via the radiative forcing (Unger et al., 2014; Ward et al., 2014) and over the long-term, via the climate-carbon feedback (Fu et al., 2020). The effect of historical or projected LULCCs on BVOCs emissions were reviewed, for instance, in Peñuelas & Staudt (2010), Unger et al. (2013, 2014) and Hantson et al. (2017). Estimates of past and future emissions accounting for climate change and increasing CO$_2$ levels rely on dynamic global vegetation models such as ORCHIDEE (Krinner et al., 2005), LPJ-GUESS (Sitch et al., 2003), SDGVM

(Woodward & Lomas, 2004) or CLM (Lawrence et al., 2019). However, human-driven land use practices must be included from other independent datasets of crops from Ramankutty and Foley (1999) or from De Noblet-Ducoudré and Peterschmitt used in Lathière et al. (2010), the 1500–2100 land use dataset from Hurtt et al. (2011) or the 2015-2100 GCAM-Demeter land use dataset from Chen et al. (2020). The evaluation of present-day impact of LULCC on BVOC emissions could however benefit from the availability of satellite observations. Remotely-sensed land cover (LC) maps are built with either discrete





classification schemes, the result of which is a raster (e.g. ESA-CCI-LC, MODIS MCD12Q1 and MCD12C1) or as continuous classification, viewing vegetation as a continuum, obtained with the use of vegetation spectral indices (e.g. MODIS VCF MOD44B; AVHRR VCF VCF5KYR; Sexton et al., 2013; Hansen et al., 2013). Traditionally, LC products use a discrete biome-based classification approach. The main drawback is that biomes are not natural vegetation units with common physiological and biochemical features required in the land surface modelling but are products of classification. Plant

functional types, or PFTs, commonly adopted in modelling, comprise plant species that share similar plant physiognomy (tree, shrub or grass), leaves (needleleaf or broadleaf), phenology (evergreen or deciduous) and photosynthetic types (C3 or C4) for crops and grasses (Smith et al., 1997; Bonan et al., 2002). Currently, PFT classifications are obtained through the mapping from biome schemes, a complex task that is flawed by arbitrariness (Bonan et al., 2002; Sun and Liang, 2008; Ustin and Gamon, 2010; Poulter et al., 2015).


The present study aims to incorporate different satellite-based land cover datasets in the MEGAN model (Guenther et al., 2012, 2016) for estimating global isoprene emissions, and to give a measure of the uncertainty associated to their use. It complements the study of Chen et al. (2018) about the impact of LULCCs on isoprene emissions based on remotely-sensed LC. The methodology is presented in Sect. 2. Section 3 reviews the distribution and trends of tree cover (TC) through 2001-2016 at

regional and global scales from different satellite-based LC products. Comparison with the latest 2020 database from Forest Resources Assessment (FRA) is made and trends over large forested regions are discussed. In Sect. 4, we perform a sensitivity analysis and quantify the impact of LULCCs on estimated isoprene emissions using MEGAN-MOHYCAN (Guenther et al., 2012; Müller et al., 2008). In Sect. 5, the IMAGESv2 global chemistry-transport (CTM) model with BVOCs emissions obtained in Sect. 4 is used to evaluate the simulated interannual variability and trends of formaldehyde (HCHO) columns

between 2005 and 2016 against satellite observations from the Ozone Monitoring Instrument (OMI). While direct satellite observations of isoprene are still in the early stages of development (Fu et al., 2019; Wells et al., 2020), the evaluation method used here based on spaceborne formaldehyde (HCHO) relies on the fact that HCHO is a high-yield product of oxidation of isoprene and has been widely used in past studies (Palmer et al., 2006; Millet et al., 2008; Stavrakou et al., 2009; Marais et al., 2012; Bauwens et al., 2016; Kaiser et al., 2018). Conclusions are drawn in Sect. 6.

**2 Methods: Datasets and models description**

**1.2 MEGAN-MOHYCAN: Biogenic VOC emission modelling**

The biogenic emissions of isoprene, monoterpenes, and 2-methylbutenol are estimated using the Model of Emissions of Gases and Aerosols from Nature (MEGAN; Guenther et al., 2006; 2012) coupled with the Model for Hydrocarbon emissions by the CANopy (MOHYCAN; Wallens, 2004; Müller et al., 2008), a multi-layer canopy environment model. MEGAN estimates the

net emission rates F ($\mu g\ m^{-2}\ h^{-1}$) into the above-canopy atmosphere using simple mechanistic algorithms encapsulated in the following equation:



$$F = \epsilon \cdot \gamma, \; with \; \gamma = \; C_{CE} \cdot \gamma_{PT} \cdot LAI \cdot \gamma_A \cdot \gamma_{CO_2} \cdot \gamma_{SM},$$

where the emission factor ε ($\mu g\ m^{-2}\ h^{-1}$) represents the emission rate at standard conditions as defined by Guenther et al. (2006). Deviations from those conditions are accounted for by the activity factors γ representing the response of biogenic emissions

to their major identified environmental and phenological drivers such as leaf temperature (T), photosynthetic photon flux density (P), soil moisture (SM), $CO_2$ concentration, leaf age (A) and leaf area index (LAI). The temperature and light response algorithm incorporates the influence of the past conditions. The adjustment factor $C_{CE}$ related to the canopy environment model is set to 0.52 for MOHYCAN, so that γ=1 at standard canopy conditions. The effects of $CO_2$ inhibition and soil moisture stress are neglected here ($\gamma_{CO_2} = 1$ and $\gamma_{SM} = 1$).


The meteorological fields are obtained from ECMWF (European Centre for Medium-Range Weather Forecasts) Interim reanalysis (ERA-Interim; Dee et al., 2011). The canopy model determines the leaf temperature and the radiation fluxes as function of height inside the canopy. The land cover features are described by the LAI and the vegetation map, classified as PFTs. Monthly LAI distributions at 0.5°×0.5° resolution (in $m^2\ m^{-2}$) are based on the MODIS dataset (MODIS 15A2H

collection 6) available at https://lpdaac.usgs.gov. Following Guenther et al. (2006) and Müller et al. (2008), we assume that the foliage covers only the vegetated fraction of the grid cell. The stand-alone MEGANv2.1 uses a static vegetation map that provides the spatial distribution of 16 PFTs for present day compatible with the Community Land Model version 4 (CLM4; Lawrence and Chase, 2007; Lawrence et al., 2011) including trees, shrubs and grasses (Table S1). In each model grid, vegetation is defined by the fractional coverage of each of the PFT. This PFT distribution was based on various satellite

products from MODIS, AVHRR, as well as the global cropland distribution for year 2000 from Ramankutty et al. (2008) and hereafter, it is referred to as CLM. The emission factor ε is calculated based on PFT-dependent emission factors provided in Guenther et al. (2012) weighted by the fractional areal coverage of the corresponding PFT class of a grid cell (Table S1).

## 2.2 Satellite-based vegetation datasets

The current satellite-based LC products cannot be directly translated into PFT classes for use in MEGAN-MOHYCAN since

they differ by their primary classification, traditionally biome-based, and by the number of classes (Sun and Liang, 2008). In order to generate MEGAN-compatible LC maps, the biome classes were first cross-walked (reclassified) into phenology-based PFT classes. This step is a cause of uncertainty due to the relative arbitrariness of the cross-walking land cover legend tables resulting from the sometimes ambiguous definitions of the biome classes. Next, the PFT trees and shrubs were further subdivided into zonal or geographical subtypes (tropical, temperate and boreal), and grasses were subdivided in photosynthetic

pathways C3 and C4, based on the Köppen-Geiger maps and climatological considerations, as explained in the following section.



| Products | Satellite sensors | Resolution | Availability |
|---|---|---|---|
| **MCD12Q1** (v006) Friedl et al. (2019) *Discrete* LULC using LCCS as PC | MODIS Terra/Aqua | 500 m | Global, gridded, annual (2001-2019) https://lpdaac.usgs.gov |
| **ESA CCI-LC** (v2.0.7 and v2.1.1) ESA CCI-LC (2017) *Discrete* LULC using LCCS as PC | AVHRR MERIS FR & RR SPOT-VGT PROBA-V | 300 m | Global, gridded, annual (1992-2019) https://maps.elie.ucl.ac.be https://cds.climate.copernicus.eu |
| **GFW** (v1.6) Hansen et al. (2013) *Continuous* TC field | Landsat & MODIS | 30 m | Global, gridded, possibility to reconstruct annual updates (2000-2019) based on the three datasets provided: (i) TC for 2000; (ii) cumulative TC gain for 2000-2012; (iii) tree loss for every year between 2001 and 2019. https://earthenginepartners.appspot.com |

**Table 1: Main features of the satellite vegetation products used for the comparison of land cover maps. Discrete LULC products,**
**namely MCD12Q1 and ESA CCI-LC, rely on a primary classification (PC) such as the LCCS, standing for Land Cover Classification**
**System. Details are found in Sect. 2.2.**

| Short name of land cover maps used in this study | Original satellite-based products |
|---|---|
| CLM | CLM4 PFT |
| ESA | ESA CCI-LC |
| MODIS | MODIS PFT (MCD12Q1) |
| GFWMOD | TC from Hansen et al. (2013) and MODIS PFT |

**Table 2: Land cover maps considered in this study, including their labels and the datasets on which they are based (described in**
**Table 1). The target period of this study is 2001-2016.**


Three datasets were considered that provide global-scale time-dependent vegetation maps over 2001-2016 (Table 1 and 2).
Those include the land cover maps based on biome classes from 1) the MODerate resolution Imaging Spectroradiometer
(MODIS), 2) the European Space Agency (ESA) Climate Change Initiative Land Cover (CCI LC), and 3) the tree cover product
of the Global Forest Watch (GFW, Hansen et al., 2013, 2020) based on 30-m Landsat images. The schematic representation
of the consecutive transformations applied on the original datasets is shown in Fig. S1.

**2.2.1 Köppen-Geiger biome types**

The subdivisions of climate zones and C3/C4 photosynthetic paths are obtained based on Table 3 from Poulter et al. (2011)
that establishes a simplified correspondence between the Köppen-Geiger classes and climate zones defined on the basis of
temperature criteria. In particular, the distinction between C3 and C4 photosynthesis adaptations is set at the threshold





temperature of 22°C (Collatz et al., 1998). The methodology is described in the Supplement. The distributions of biomes from
       the original classification of Poulter et al. (2011) and the modified version thereof are listed in Table S2 and displayed in Fig.
       S2. We use the 0.5°×0.5° resolution, global Köppen-Geiger present-day (1980–2016) climate classification map from Beck et
       al. (2018), developed at 1-km resolution. The Köppen-Geiger maps are available at http://www.gloh2o.org/koppen.

### 2.2.2 ESA-CCI LC: Land cover map

The ESA-CCI-LC product supplies global annual land cover maps at 300-m resolution (ESA-CCI-LC, 2017). Maps were
       generated by combining the global daily surface reflectance of five different observation systems: Advanced Very High
       Resolution Radiometer (AVHRR), Satellite Pour l'Observation de la Terre - Vegetation (SPOT-VGT), PRoject for On-Board
       Autonomy - Vegetation (PROBA-V), and Medium Resolution Imaging Spectrometer (MERIS) Full and Reduced Resolution
       (FR & RR).

       The original product has 37 classes from the UN-LCCS. The cross-walking table for their conversion to PFTs is taken from
       Li et al. (2018) based on Poulter et al. (2015). The PFT mapping and the aggregation into 0.5°×0.5° were performed using the
       ESA-CCI-LC user tool (version 4.3) available at https://maps.elie.ucl.ac.be. The mapping to climate zones and photosynthetic
       paths is applied as described above. The final land cover map will be referred to as ESA.

### 2.2.3 MODIS PFT: Land cover map

The collection 6 of the MODIS Land Cover Type Product (MCD12Q1; Friedl et al., 2019) provides a global land cover product
       at 500-m resolution at yearly intervals from 2001 to present. The product is derived from the reflectance data of the Terra and
       Aqua missions. We use here the MODIS LC product available with the PFT legend. Unlike in collection 5, where 17 IGBP
       (International Geosphere–Biosphere Programme) classes were cross-walked to create annual maps for the PFT scheme, the
UN-LCCS (United Nations Land Cover Classification System (Di Gregorio, 2005) scheme provides the primary layer for the
       sixth collection. Note however that the corresponding cross-walking table is not released.  The maps are aggregated to
       0.5°×0.5°, on which further subdivisions of climate zones and photosynthetic paths are applied following the methodology
       described above.

### 2.2.4 GFW: Tree cover map

Unlike the discrete LC maps of MODIS PFT and ESA-CCI-LC, the Global Forest Watch dataset (Hansen et al. 2013; 2020)
       is a continuous field of tree cover (TC) coverage, available at a global scale with approximately 30-m resolution. In recent
       years, it has become a reference for monitoring forests through the online platform Global Forest Watch
       (www.globalforestwatch.org). It is generated by combining data from the multispectral sensors Landsat 5 and 7 for 2000-2012
       and Landsat 8 for 2013 onward. The resulting images are normalised by using MODIS surface reflectance (Hansen et al., 2008;
Potapov et al., 2012). The version 1.6 used in this study covers all years from 2000 to 2018, and provides tree losses on an





annual basis over 2000-2018 and a cumulative tree gain distribution over 2000-2012. We reconstructed annual steps of the TC at pixel level (30-m) by accounting for the TC losses since 2000, and by implementing the 12-year cumulative tree cover gain 0.5° resolution assuming a linear increase over the period, extended until 2018. The land cover distribution GFWMOD (Table 2) is obtained by modifying the MODIS PFT dataset in order to match the reconstructed yearly TC distribution based on the
GFW dataset.

### 2.3 Forest Resources Assessment (FRA) database – Food and Agriculture Organization (FAOSTAT)

The standard reference on global forest resources is the United Nations FAO FRA published every 5 to 10 years since 1948. It provides the global database of the reported statistics from national reports on forest properties. The latest recommendations are outlined in the FRA 2020 (http://www.fao.org/3/I8661EN/i8661en.pdf). Here we use the latest published datasets (updated
in July 2020 at the time of drafting), retrieved from http://www.fao.org/faostat, and hereafter referred to as FAOSTAT.

### 2.4 Formaldehyde observations from OMI

Tropospheric HCHO columns are acquired from satellite-based OMI observations on board the NASA AURA spacecraft launched in 2004 (Levelt et al., 2006). OMI is a nadir-viewing imaging spectrometer observing Earth's global solar backscatter radiation in the ultraviolet-visible spectral window at a spectral resolution of about 0.5 nm. It has an early afternoon overpass
time (13:30 local time) and provides global observation on a daily basis at a spatial resolution of 13 x 24 km² at nadir.
The OMI HCHO product used in this study was developed in the framework of the EU FP7 project QA4ECV (Quality Assurance for Essential Climate Variables, http://www.qa4ecv.eu) and is documented in De Smedt et al. (2015; 2017; 2018). The retrieval approach consists of three steps. Firstly, the HCHO slant columns are retrieved in the 328.5-359 nm spectral window using up-to-date Differential Optical Absorption Spectroscopy (DOAS) algorithms (De Smedt et al., 2018) with the
HCHO absorption cross-section from Meller and Moortgat (2000). Secondly, for weak absorbers such as HCHO, the background normalisation of the slant columns using the equatorial Pacific Ocean as the reference sector is applied in order to compensate for possible systematic latitude-dependent offsets in spectral fitting. Eventually, the conversion into a vertical column is performed with the air mass factor (AMF) assuming an optically thin approximation. The latter is obtained from an altitude-resolved AMF look-up table derived at 340 nm from the VLIDORTv2.6 radiative transfer model (Spurr et al., 2008)
and the daily a priori vertical profile shape of the HCHO distribution calculated with the TM5-MP chemical transport model (Huijnen et al., 2010; Williams et al., 2017). The scattering due to clouds is corrected using the independent pixel approximation (Martin et al., 2002), whereas an implicit correction for the effects of aerosols is accounted for through the cloud correction scheme.

### 2.5 The IMAGESv2 CTM

Simulations of the atmospheric composition are performed using IMAGESv2 (Intermediate Model of the Global and Annual Evolution of Species), a global three-dimensional CTM of the troposphere (Müller and Stavrakou, 2005; Bauwens et al., 2016;

Stavrakou et al., 2016; 2018). The model calculates the concentrations of 170 compounds in the global troposphere with a spin-up time of six months. The horizontal resolution of the model is 2°×2.5°, while the vertical coordinate is a hybrid sigma-pressure system resolved in 40 unevenly spaced levels extending from the surface to the lower stratosphere (44 hPa). The

model is driven by ERA-Interim meteorological fields (Dee et al., 2011). The chemical degradation mechanism is described in Bauwens et al. (2016) and Stavrakou et al. (2018).

The bottom-up fluxes of HCHO precursors are prescribed as follows. The biomass burning inventory is provided by the Global Fire Emissions Database version 4s (GFED4s; van der Werf et al., 2017) on a daily basis and a global spatial resolution of

0.25°×0.25° (https://globalfiredata.org). The anthropogenic sources of NMVOC species are taken from the EDGARv4.3.2 (Emission Database for Global Atmospheric Research; Huang et al., 2017) and the anthropogenic $NO_x$, CO, $SO_2$ and $NH_3$ are obtained from the HTAPv2 (Hemispheric Transport of Air Pollution; Janssens-Maenhout et al., 2015) database for 2010. Due to their limited availability, the EDGARv4.3.2 emissions are set constant at their 2012 values after this year. Both inventories are available at https://edgar.jrc.ec.europa.eu. The biogenic emissions of isoprene, monoterpenes, and methylbutenol are

estimated as described in Sect. 2.1. The biogenic emissions of acetaldehyde and ethanol are parameterized following Millet et al. (2010). Biogenic CO emissions and CO deposition are accounted for following Müller and Stavrakou (2005). Finally, the biogenic methanol emissions are provided by an inverse modelling study constrained by spaceborne methanol data (Stavrakou et al., 2011).

### 3 Comparison of satellite-based tree cover datasets

The LULC datasets shown in Table 2 are compared in detail below over the 2001-2016 period, namely MODIS, GFWMOD, and ESA.  In this study, the tree cover refers to the aggregation of the 8 PFTs corresponding to trees from the CLM4 PFT classification scheme (Table S1).

### 3.1 Global TC areas, spatial distributions and trends

The global TC distributions for year 2001 are depicted in Fig. 1. The global TC areas provided therein and listed in Table 3

range from 30.6 Mkm² for ESA to 52.6 Mkm² for MODIS, with the TC areas of GFWMOD and CLM falling in between with 32.2 and 38.5 Mkm², respectively. MODIS TC stands out as it exhibits extensive patches of high TC densities (> 90%) in all major forested regions. In contrast, GFWMOD and ESA exhibit lower densities (40-80%) in the Northern Hemisphere. The ESA cover density reaches 90% in the tropical forests of Central Africa, Amazon, Southeast Asia and Oceania. The lowest densities are found in the ESA dataset, e.g. in the Eastern U.S., the West African coast and Indonesia.


Table 3 compares the global satellite-based TC with figures from FAOSTAT based on national reporting. According to FAOSTAT, the global forest area reached 41.5 Mkm² in 2001, which lies well within the satellite-based TC areas.





| | Area in 2001 | Trends | |
|---|---|---|---|
| | (Mkm²) | (% yr⁻¹) | (km² yr⁻¹) |
| **FAOSTAT** | 41.5 | -0.12 | -49,727 |
| **CLM** | 38.5 | --- | --- |
| **ESA** | 30.6 | -0.05 | -13,968 |
| **MODIS** | 52.6 | 0.03 | 18,184 |
| **GFWMOD** | 32.2 | -0.26 | -83,336 |

**Table 3: Global TC areas (in Mkm²) in year 2001 and trends (in % yr⁻¹ and in km² yr⁻¹) over 2001-2016 based on FAOSTAT, the static land cover map (CLM) and the satellite products (ESA, MODIS and GFWMOD).**

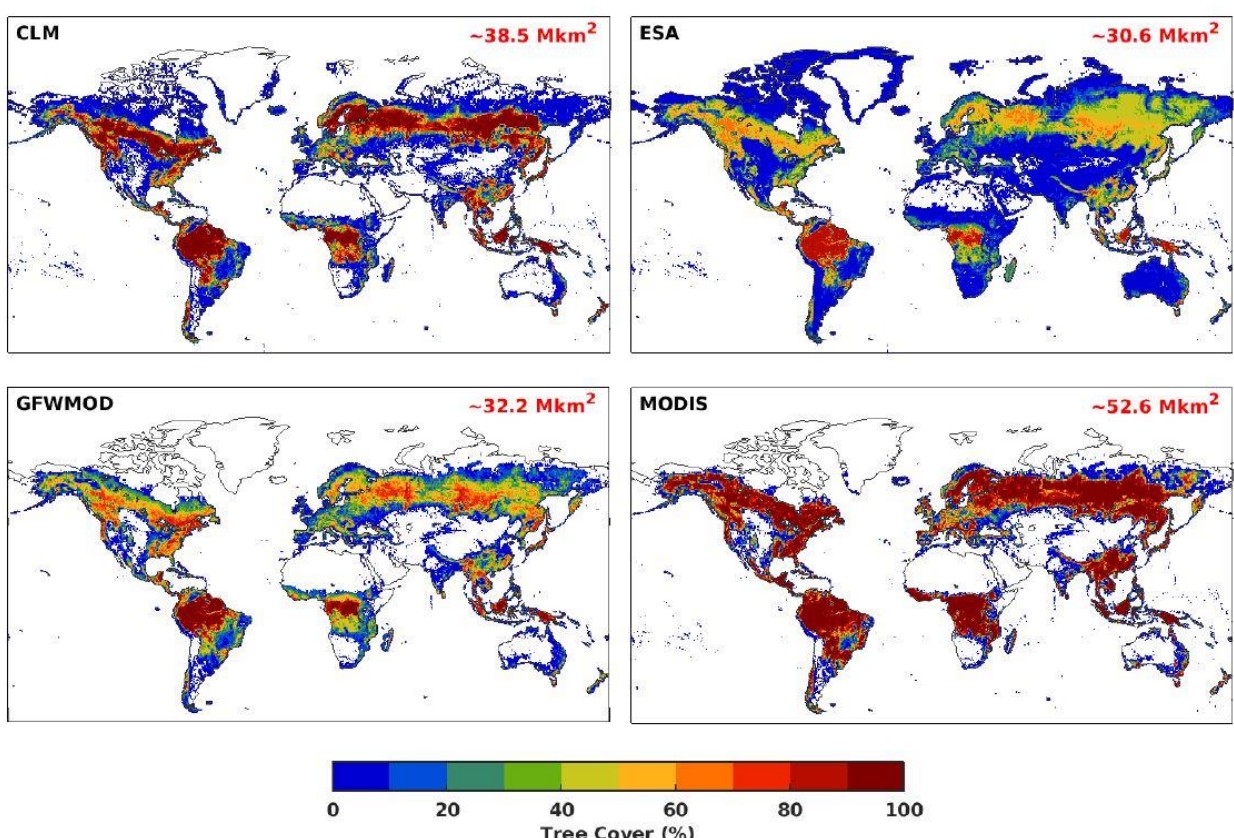

**Figure 1: Spatial distribution of the TC for the static land cover map CLM (for the present day) and for the satellite-based datasets (ESA, MODIS, GFWMOD) for year 2001. The corresponding global TC areas are provided inset.**

Time series of global TC areas normalised to the 2001 values are displayed in Fig. 2(a) and total global trends are shown in Fig. 2(b) and Table S2 for 2001-2016. Of all datasets, GFWMOD exhibits the strongest negative trend, equal to –0.26% yr⁻¹




(ca. -83,500 km² yr⁻¹), which is about 3-5 times as fast as the FAOSTAT and ESA trends (Table 3). The ESA dataset shows the lowest variation (ca. -14,000 km² yr⁻¹), with a stable phase between 2004 and 2009. While net deforestation is found in both GFWMOD and ESA datasets, the MODIS dataset exhibits a small positive linear trend of ~0.03% yr⁻¹. The MODIS TC

declines in 2001-2003 and after 2014 are more than compensated by the slow increase in 2003-2014 (Fig. 2(a)).

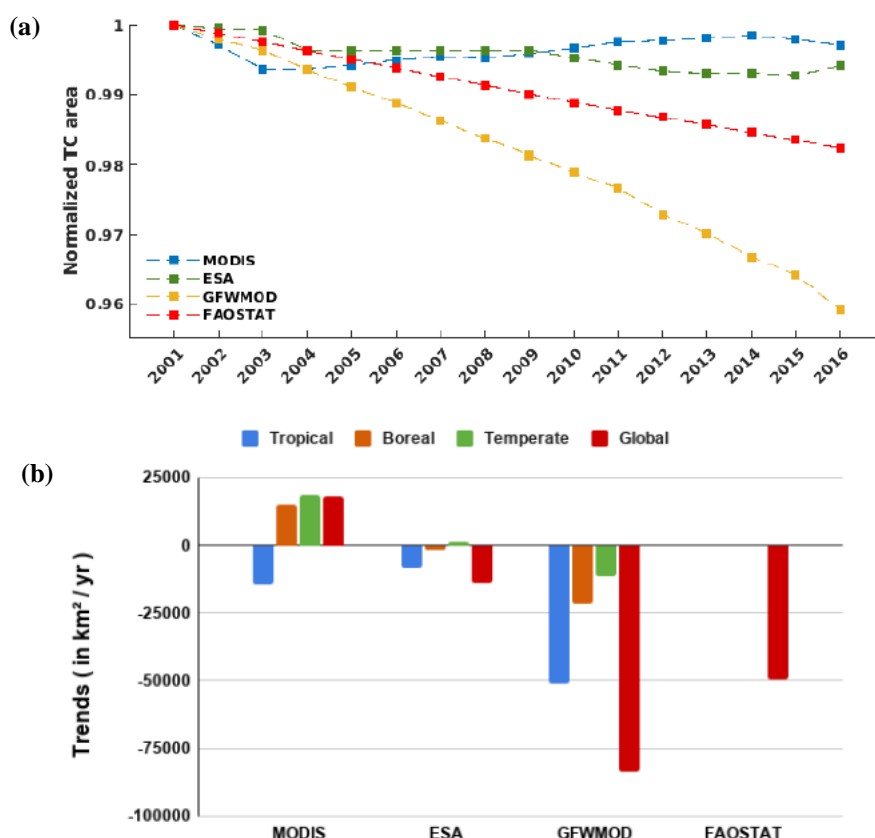

**Figure 2: TC trends over 2001-2016. (a) Time series of the global net changes of TC areas (normalized to 2001 values) from FAOSTAT, MODIS, ESA and GFWMOD datasets, and (b) Total net TC trends at global scale and per climate zone (as defined in Sect. 2 of the Supplementary Material and depicted in Fig. S1(b)). For the FAOSTAT dataset, TC trends per climate zone are not available.**

Of all biomes, tropical trees experienced the greatest net losses according to GFWMOD and ESA datasets. This loss is 3.5 times greater in GFWMOD than in ESA (Fig. 2(b) and Table S3). MODIS also presents a net decline in tropical trees (-14,600 km² yr⁻¹), almost entirely located in South America (Fig. 3) with little net changes over Africa and South-East Asia, but it is

offset by positive trends in the boreal and especially in the temperate domain. Unlike GFWMOD, which features a net loss in all domains, MODIS demonstrates net gains at mid- and high latitudes of the Northern Hemisphere, with the biggest changes encountered in temperate forests (18,400 km² yr⁻¹). As a result, these biomes, with strong trends and covering together roughly



60% of the total TC area, drive the global net increase found in MODIS dataset. Compared to the other two datasets, the MODIS distribution shows significant impacts over much larger areas, mainly in the periphery of major forests of South America, Africa and Southeastern U.S. In contrast, GFWMOD displays large net changes within higher-density forest canopies like those in the Tropics, Southeastern U.S., China, and Scandinavia. In the ESA dataset, the net TC changes are sparse and weak, about one order of magnitude lower than in the two other satellite-based products, and show highest values in boreal and temperate regions (Fig. 3).

### 3.2 National and regional distributions of trends in large forested countries

The satellite-based TC trends are compared against national inventories collected through FRA national reports for several large countries in Figure 4 and Table S4. Overall, large discrepancies are found across the different estimates with respect to both the magnitude and the sign of trends. The regional differences are further discussed based on Fig. 3.

### 3.2.1 United States of America

The U.S. FRA report indicates a positive trend of 4,500 km² yr$^{-1}$ over the period considered (2001-2016), whereas the satellite-based records suggest a net declining TC area. According to GFWMOD, Eastern U.S. forests experience a net deforestation, except in parts of Louisiana, Mississippi, and Florida (Fig. 3). Both MODIS and ESA show little net changes in the Eastern U.S., except in the Mississippi Alluvial Plain. In the northwest, forests undergo deforestation according to all datasets. In Alaska, GFWMOD indicates a clear net loss over the Yukon River Basin, also seen in the ESA dataset although at lower rates. Overall, MODIS shows strong positive trends in areas of low TC densities.

### 3.2.2 Brazil and Indonesia

There is qualitative agreement among all datasets regarding the TC trends in Brazil and Indonesia, even though there are large differences in absolute terms. Over Indonesia, the FAOSTAT trend (-4,000 km² yr$^{-1}$) lies within the estimates based on ESA (-2,550 km² yr$^{-1}$) and GFWMOD (-7,250 km² yr$^{-1}$), whereas very little changes are suggested by the MODIS dataset (-830 km² yr$^{-1}$). In Brazil, all satellite-based estimates underestimate the national inventory (-32,200 km² yr$^{-1}$), by factors of about 2-3 in the cases of GFWMOD (-16,900) and MODIS (-10,600), while ESA differs from the other satellite-based products with very low and sparse net total loss (- 930 km² yr$^{-1}$). GFWMOD exhibits deforestation across the Cerrado and Caatinga regions, with the border regions of the Amazon forest-Cerrado experiencing the largest net losses. In MODIS, strong positive and negative trends are found in areas of the low TC density areas contiguous to Paraguay and Caatinga, respectively (Fig. 1). The southern part of the Atlantic Forest along the eastern coast of Brazil experiences net TC increases according to GFWMOD and ESA, but not MODIS.



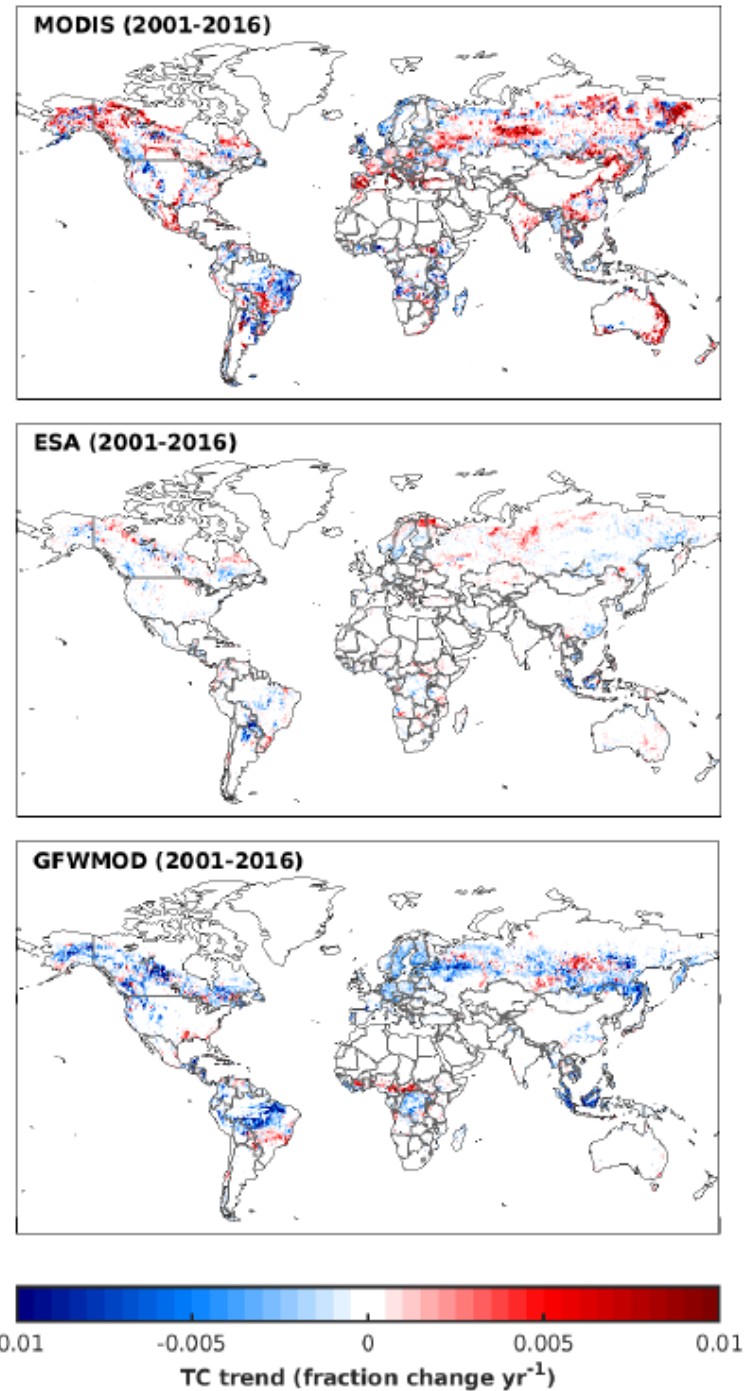

**Figure 3: Spatial distribution of the linear net trend (change in cover fraction per year) in TC for 2001-2016 in the MODIS, ESA, and GFWMOD datasets. The fraction change per year is calculated by dividing the TC trends in each grid cell expressed in km² yr⁻¹ by the corresponding grid area.**



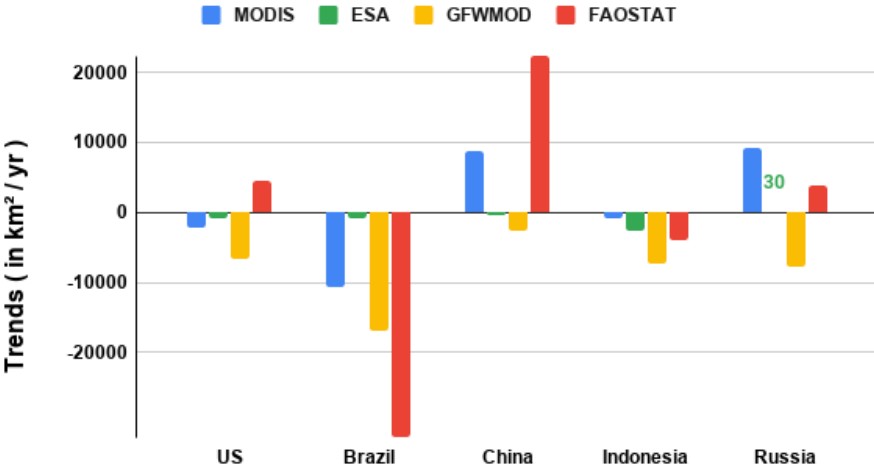

**Figure 4: Net total TC trends (in km² yr⁻¹) in five large countries according to MODIS, ESA, GFWMOD, and FAOSTAT datasets for 2001-2016.**

### 3.2.3 China

The strong positive trend in forest cover (~22,000 km$^2$ yr$^{-1}$) reported by FAOSTAT in China is not supported by any space-based estimate. MODIS indicates a positive, but much smaller trend (+8,800 km² yr$^{-1}$), whereas both the GFWMOD and ESA datasets exhibit a small net loss of about -2,500 and -350 km² yr$^{-1}$, respectively. The spatial TC distributions reveal trends mostly in the southern part of the country. In the Yunnan-Guizhou Plateau, MODIS exhibits net positive trends, unlike ESA and GFWMOD that show null net trends. Guangdong also experiences a net increase in TC according to MODIS, whereas both GFWMOD and ESA indicate significant declines in Southeastern China in line with the strong deforestation found in these datasets.

### 3.2.4 Russia

Over Russia, FAOSTAT and MODIS record total net increasing trends of 4,000 and 9,300 km² yr$^{-1}$, respectively, while the trend is negligible for ESA, and GFWMOD presents a net negative trend, ca. –7,700 km² yr$^{-1}$. All satellite-based trend distributions (Fig. 3) exhibit net positive trends in a large zone around 55°N between Belarus and the West Siberian Plains. In the Central Siberian Plateau, the strongest net positive and net negative trends of GFWMOD are respectively located north and east of Lake Baikal. GFWMOD differs from the other datasets in that the trend patterns are uniformly distributed across the forested regions, whereas trends found in ESA and MODIS lie mainly on the outskirts thereof. Over northeastern Siberia, MODIS shows a net increase in the forested area, located close to the Kamchatka peninsula (Fig. 1), contrasting with the negative tendencies found with the ESA and GFWMOD products over the same region.



### 3.3.  Reasons for disparities in tree cover areas and trends

The paramount difference between all products is the definition of the tree or forest cover. A height threshold allows to separate a tree from a shrub. Usually, woody plants higher than 5 m are classified as trees. A forest cover is derived from the tree cover by applying a minimum threshold for the canopy cover and integrating over entire pixel areas where the condition is met. Table 4 summarizes the criteria of height and canopy density defining the TC/FC of all datasets of the study.


| | PFT TC definition | | Spatial resolution (m) |
|---|---|---|---|
| | TC Heights (m) | Canopy density (%) | |
| FAOSTAT (OFFICIAL) | > 5[a] | > 10[a] | 71[a] |
| MODIS | > 2 | > 10 | 500 |
| ESA | > 5[b] | > 15 | 300 |
| GFWMOD | > 5 | N/A[c] | 30 |

**Table 4: Criteria for the classification of tree PFTs as inherited from the different LULC datasets (Table 1). [a] Note that the FAOSTAT definition is based on the FRA 2020 recommendations but is not consistently applied in national reports. [b] This general rule is subject to an exception and accounts for trees with height > 3 m if a clear physiognomic aspect of trees is detected. [c] No**
**threshold is assumed for GFW.**

As a first instance, the disparities in TC/FC between earth observation-derived products and the FAOSTAT database can, to a large degree, be attributed to differences in the definitions used. According to the FRA report, a forest is defined by a minimum threshold of 5 m height, a canopy closure of minimum 10%, and a minimum area cover of 0.5 ha, that is a parcel of ca. 71×71 m² which includes trees able to reach these thresholds (FRA 2020 'Terms and Definitions',
http://www.fao.org/3/I8661EN/i8661en.pdf). This definition is tailored to a forest land use description but is ill-suited from a biophysical perspective. As long as they meet the biophysical criteria with respect to canopy density and/or height, human managed lands such as rubber plantations and agroforestry are construed as trees in the satellite imagery data, whereas the FRA excludes land use covers with extended human interference such as agricultural and urban land use.  Besides, the national reports often rely on methodologies, which are not in accordance with FRA recommendations. The plurality of definitions in
use and the associated issue of directly evaluating satellite-based dataset against FAOSTAT database has been pinpointed in previous research (Hansen et al., 2013; GFW article; Li et al., 2018; Nomura et al., 2019).

The FAOSTAT numbers for U.S., China and Russia rely on field work inventories. For the U.S., the official reporting is provided by the Forest Inventory and Analysis Program of the U.S. Forest Service that classifies clear cut forest as well as
seedling and young trees as forest (FRA national report 2020: http://www.fao.org/3/cb0086en/cb0086en.pdf). National reports





of China are based on a definition of forest cover that accounts for a great variety of vegetation falling in the forest class, which is defined as an area spanning more than 0.0667 ha with a canopy density above 20% (FRA national report: http://www.fao.org/3/ca9980en/ca9980en.pdf). Nursery land and cut- and fire-over areas that do not meet the biophysical requirements stated by the FAO user guide are included as well as economic and bamboo forests. The inclusion of seedling and young trees could be swelling the positive trends of U.S. and China reports. The 2020 FRA report of the Russian Federation is based on the State Forest Inventory provided by the Russian Research Institute for Silviculture and Mechanization of Forestry in Moscow. According to the FRA report for Russia (FRA national report: http://www.fao.org/3/cb0053en/cb0053en.pdf), almost 80% of the total land area on which forests are located in "hard-to-reach" beyond the 60th parallel (Alekseev et al., 2019). Since most of the deforestation caused by wildfires takes place in those regions (Curtis et al., 2018), inventories could present large underestimations of the reported losses. Unlike other countries, both Brazil and Indonesia use Landsat imagery to estimate the forest changes provided in the FRA reports, which could explain the better qualitative match between the satellite-based trends and the FRA figures, whereas discrepancies in their magnitude might be due to differences in classification forest/non-forest (FRA national report of Brazil: http://www.fao.org/3/ca9976en/ca9976en.pdf and Indonesia: http://www.fao.org/3/cb0007en/cb0007en.pdf).

The comparison of the three satellite-derived products has also shown great discrepancies in their spatial distributions, areas and trends. Those differences stem from various factors leading to uncertainties and inconsistencies: acquisition methods (e.g. missions and sensors; Table 1), mapping methodology (classification algorithms of spectral reflectance into LC classes), original classification definition related to height and canopy thresholds, the conversion of the original LC into PFTs distribution (Congalton et al., 2014), as well as the spatial resolution.

Although all datasets were mapped to the 16 CLM4 PFTs required for MEGAN-MOHYCAN (Table S1), it is important to stress that the definitions of LC classes differ among the datasets. In particular, the criteria defining the tree PFTs (Table S1 and Table 4) are critical. On one hand, the tree PFTs in discrete LULC products (MODIS and ESA) are defined by a minimal threshold of canopy cover and hence, represent a forest cover instead of a tree cover. On the other hand, the GFW dataset provides continuous TC fields without involving any a priori threshold on canopy density. This difference might account for a part of the differences between the MODIS- and ESA-based estimates (Fig. 3) with respect to GFWMOD, since there can be a substantial change in tree density without implying a change in FC. The threshold on canopy cover affects considerably the areal tree cover and net trends. For instance, the GFW-based TC area in 2000 would amount to ca. 40 Mkm² if the canopy density threshold of >25% were applied at the Landsat pixel scale (30 x 30 m²) (Hansen et al., 2013). Here, we do not apply any threshold and simply average the TC densities from GFW onto 0.5°×0.5° grid cells. For this reason, the TC areas reported for the GFWMOD dataset differ from studies that used the minimum threshold of 25% (Li et al., 2018; Hansen et al., 2013). The global net trends from GFWMOD are the largest among all datasets. Since it is the actual TC that matters in the biosphere-atmosphere exchanges of biogenic VOCs, the calculation of tree cover losses at pixel level should account for the actual





percentage changes in a given pixel. This differs from the approach of Hansen et al. (2013) according to which the forest losses
       are calculated based on the entire area of pixels reported as lost.

       Furthermore, part of the bias in the magnitude and trends of the MODIS- and ESA-based estimates with respect to GFWMOD
       stems from the PFT mapping. The often vague or inadequate definition of LC classes in the original products results in much
arbitrariness in the cross-walking tables applied to map those land cover products onto the CLM4 PFTs. As seen in Fig. 1,
       MODIS shows large areas of canopies with very high densities (> 90%), which are due to the mapping of the MODIS classes
       'sparse forest' (defined at 10–30% canopy closure) and 'open forest' (defined at 10-60% canopy closure) onto the tree cover
       PFTs (Sulla-Menashe et al., 2019). Li et al. (2018) adopted a less radical redistribution of the ESA LC classes, leading to
       generally lower densities.


       The spatial resolution plays also a role in the magnitude of changes, since finer resolutions can capture disturbances occurring
       at finer scales. GFWMOD based on 30-m pixels exhibit stronger net changes in the forested areas whereas in lower resolution
       datasets, ESA (300-m) and MODIS (500-m), the trends are representative of dominant land cover changes that are mainly seen
       at the outskirts of the forested regions, in particular in South America. The ESA dataset shows the lowest net changes because
the LC changes were first detected at 1-km resolution and then, delineated to a higher resolution of 300 meters (ESA-CCI-LC,
       2017). The fine resolution of the Hansen et al. (2013) database is a unique asset for tracking land cover changes and trends.
       However, GFWMOD comes with its own shortcomings. It inherits uncertainty and inconsistencies in the trend due to the
       changes in mapping methodology of TC losses from year 2011 onwards in the Hansen et al. (2013) dataset. The global forest
       losses of 2011-2018 used an updated processing for detection, and, at the time of drafting, the dataset was not yet reprocessed
prior to 2011.

       **4 Comparison of isoprene emissions for different satellite-based LC products**

       We investigate the effects of LULC variations on global biogenic isoprene emissions using results from three simulations:
       CTRL, using the static CLM map; ISOPMOD, using the MODIS dataset; and the ISOPGFW, using the GFWMOD dataset.
       Given the very low net changes and variability seen in the ESA dataset, the latter was discarded from further analysis. The
three simulations account for the same meteorology but differ in the input of the vegetation maps. The influence of soil moisture
       stress and $CO_2$ inhibition are neglected here, that is, $\gamma_{SM} = 1$ and $\gamma_{CO_2} = 1$ (Sect. 2.1), unless stated otherwise. The MEGAN-
       MOHYCAN model does not represent the interplay between the vegetation land cover map and meteorological conditions as
       it is the case in dynamic ecosystem models. The effects of climate and vegetation are decoupled and only direct impacts thereof
       are considered. The interannual variability of meteorological conditions is considered through ERA-Interim reanalyses,
allowing a validation with atmospheric formaldehyde to be performed in the following section.



### 4.1 Global and total emissions, distributions and trends

The average annual global isoprene emission is estimated at ca. 420 Tg for the 2001-2016 period in the CTRL simulation (Table 5), whereas it is 24% higher in the ISOPMOD run, and 15% lower in the ISOPGFW simulation. These figures remain in the range (350-800 Tg yr$^{-1}$) found in previous estimations with MEGANv2 using various drivers (Guenther et al., 2012). By

comparison, bottom-up inventories such as CAMS-GLOB-BIOv1.1 (Granier et al., 2019; Sindelarova et al., 2014), its predecessor MEGAN-MACC (Sindelarova et al., 2014) and the BIRA-IASB inventory (Stavrakou et al., 2018; https://emissions.aeronomie.be) estimate the mean global annual total of emitted isoprene in the range of 380-590 Tg. The effects of $CO_2$ inhibition and soil moisture were neglected in those inventories except for MEGAN-MACC, which accounted for $CO_2$ inhibition (Sindelarova et al., 2014). The inclusion of the $CO_2$ inhibition effect according to the parameterization of

Possell and Hewitt (2011) in our study would lead to an annual emission decrease of the order of 3% (Table S5). Including the soil moisture stress effect ($\gamma_{SM}$) based on ERA-Interim soil moisture data would lead to a further decrease of the order of 10% (Table S5). Disparities among the inventories are primarily attributed to differences in meteorological fields and emission potential distributions (Arneth et al., 2011), with additional possible contributions of differences in the canopy environment models and in the LAI datasets.


The distribution of isoprene emissions in the three simulations (Fig. 5) shows that the highest values are found over Amazonia, the Yucatan peninsula, West and Central Africa, and South-East Asia, in particular Borneo. These spatial patterns reflect the warm temperature, high radiation fluxes and high isoprene emission factors generally found in the Tropics. Secondary maxima are found at temperate latitudes during summertime, in particular over the Southeastern U.S. and Southern China. ISOPMOD

predicts higher emissions than the other two simulations over many regions, in consequence of its higher tree cover (Fig. 5 and Fig. S3), in particular over northeastern Brazil, large parts of Africa, South China and Eastern U.S. Regional dissimilarities between the datasets (Fig. S3) are mainly located in isoprene-rich areas and stem from differences in the canopy coverage. In some regions (e.g. North-West Australia), both ISOPMOD and ISOPGFW exhibit higher emissions compared to CTRL due to their higher shrub extent, with shrubs being relatively high emitters. Since the GFWMOD dataset inherited PFTs (besides

the total tree cover) from the MODIS PFTs dataset, the spatial patterns of ISOPGFW and ISOPMOD are similar in low-TC areas.

| | Mean (in Tg) | Trend (% yr$^{-1}$) | max IAV (%) |
|---|---|---|---|
| **CTRL** | 418 | 0.94 | 20 |
| **ISOPMOD** | 520 | 0.90 | 19.5 |
| **ISOPGFW** | 354 | 0.61 | 18 |

**Table 5: Global mean annual isoprene emissions (in Tg), trend (% yr$^{-1}$) and maximum interannual variability (IAV, defined as difference between global maximum and global minimum, in %) for the 2001-2016 period in CTRL, ISOPMOD and ISOPGFW**
**simulations.**





**Figure 5: Distribution of isoprene emissions (in mg m⁻² day⁻¹) for June-July-August (JJA, left panels) and December-January-February (DJF, right panels) in 2001 in simulations CTRL (upper row), ISOPMOD (middle row) and ISOPGFW (lower row).**



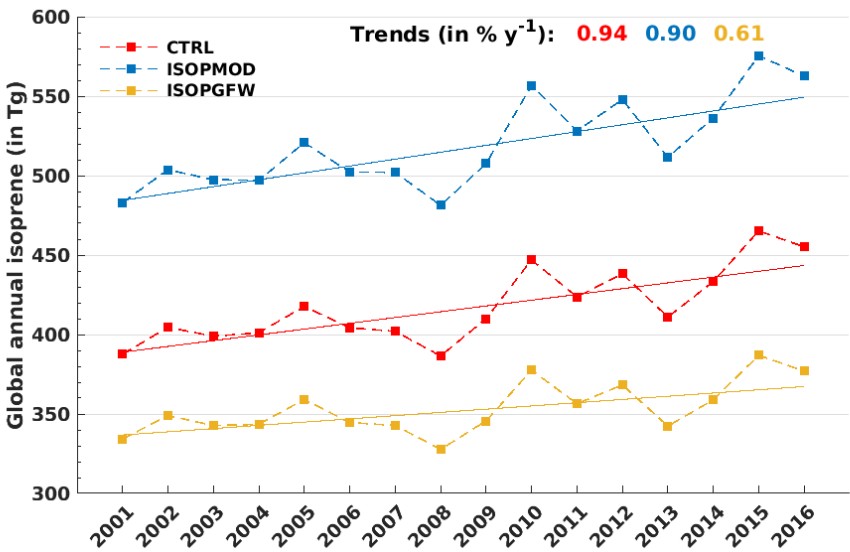

**Figure 6: Time series of global annual isoprene emissions from the CTRL, ISOPMOD and ISOPGFW simulations.**

The trend of the global annual isoprene emissions in the CTRL run is estimated at 0.94% yr$^{-1}$ (Fig. 6). Since the PFT distribution and therefore the isoprene emission factors were held constant in this simulation, the interannual variability of emissions in this simulation is essentially due the interannual variability of meteorological parameters, primarily temperature and visible radiation fluxes. Accounting for LULC changes with the MODIS and GFWMOD land cover maps results in a cutback of the global isoprene emissions trends by 0.04% yr$^{-1}$ and 0.33% yr$^{-1}$, respectively. In ISOPMOD, the deceleration in the global trends pertains to a negative trend of tropical tree cover in MODIS (Fig. 2) which more than compensate for the increasing coverage of temperate and boreal trees, given the large share of the Tropics (80%) in the global emissions. In ISOPGFW, the significant decline in tree cover in all climate zones explains the strong drop in the global isoprene emission trend, from 0.94 to 0.61% yr$^{-1}$. Note that the $CO_2$ inhibition effect, not considered in those simulations, would further offset global trends by about 0.22% yr$^{-1}$, whereas the soil moisture stress has little impact on trends (Table S5). The effect of LAI trends on global emission trends is very small: a positive increment of +0.06% yr$^{-1}$ was calculated based on an additional sensitivity simulation in which LAI interannual variability was omitted.

The interannual variability (IAV) of isoprene emissions is mainly driven by meteorology and is positively correlated with the Oceanic Niño Index (ONI) (Naik et al., 2004; Lathière et al., 2006; Müller et al., 2008; Stavrakou et al., 2014). Maxima in isoprene emissions correlate with El Niño events (2002/2003, 2004/2005, 2009/2010 and the 2014-2016), and minima with La Niña episodes in 2007-2009 (NOAA ONI v5, https://origin.cpc.ncep.noaa.gov). This variability is weakly dependent on variations in PFTs. Besides the differences between long-term trends of the emissions from the three simulations, the year-to-year relative changes in emissions are very similar (Fig. 6). The global minimum and maximum occur in all datasets





respectively in 2008 and 2015, whereas the maximum IAV, defined as difference between global maximum and global minimum (in %), amounts to 20%, 19.5% and 18% in simulations CTRL, ISOPMOD and ISOPGFW, respectively (Table 5). The impact of interannual variability of LAI on the variability of isoprene emissions is considered to be either minor (Müller et al., 2008; Stavrakou et al., 2014) or uncertain, because of the large disparities in the long-term evolution of LAI products (Jiang et al., 2017).


The isoprene emissions trends of the CTRL run illustrated in Fig. 7 reflect the temperature and solar radiation trends (Fig. S4 and S5). The ability of MEGAN-MOHYCAN to reproduce the response of biogenic emissions to short-term climate variability over vegetated areas was demonstrated in Stavrakou et al. (2018) using spaceborne formaldehyde observations. The simulations ISOPMOD and ISOPGFW account for the impact of both climate variability and LULC changes. The effect of

LULC changes on emission trends can be estimated from the differences in trend between those simulations and the CTRL run. The differences, namely ISOPMOD-CTRL and ISOPGFW-CTRL, are shown in the middle and bottom panels of Fig. 7, respectively. In isoprene-rich areas, the trend pattern of ISOPMOD-CTRL and ISOPGFW-CTRL largely reflect trend in tree coverage (Fig. 3). This is because high-emitting broadleaf trees are generally dominant in those areas. In less forested regions (low TC), the spatial distribution of trends of ISOPGFW and ISOPMOD is similar since non-tree PFTs in GFWMOD and

MODIS are similar. The decreasing trends in grass and especially shrub cover in low-emissions areas of, for instance, Central US, South-East Australia and Western India are responsible for the significant relative decreases in isoprene emissions in those areas (left panels in Fig. 7).


| | Annual emissions (in Tg) | | | Annual trends (in % yr⁻¹) | | |
|---|---|---|---|---|---|---|
| | CTRL | ISOPMOD | ISOPGFW | CTRL | ISOPMOD | ISOPGFW |
| **US** | 14.5 | 24 | 16 | 1.53 | 1.51 (-0.02) | 1.27 (-0.26) |
| **Brazil** | 82 | 106.5 | 79.5 | 0.99 | 0.68 (-0.31) | 0.46 (-0.53) |
| **China** | 12 | 23 | 9.5 | 0.74 | 1.06 (+0.32) | 0.31 (-0.43) |
| **Indonesia** | 30.5 | 32 | 24.5 | 1.19 | 1.07 (-0.12) | 0.50 (-0.70) |
| **Russia** | 6 | 8 | 7 | 0.57 | 1.33 (+0.76) | 0.91 (+0.34) |

**Table 6: Annual isoprene emission estimates (in Tg) averaged over 2001-2016 in largely forested countries and isoprene trends over 2001-2016 (in % yr⁻¹). In the CTRL study, isoprene trends pertain to changes in meteorological conditions. The combined effect accounting for both meteorological and LULC trends is given for ISOPMOD and ISOPGFW while the mitigating effect from LULC changes is provided in brackets.**



**Figure 7: Global distribution of annual isoprene emissions trends for the 2001-2016 period. The upper panel displays the relative trend (% yr$^{-1}$) in the CTRL run. The lower left panels represent the relative trend differences between the runs. The lower right panels display the absolute trend differences in mg m$^{-2}$ h$^{-1}$ yr$^{-1}$.**




Of the largely forested countries shown in Table 6, Brazil and Indonesia have the highest emissions, of the order of 80 and 30 Tg yr$^{-1}$, respectively. Emissions from Russia are low, mainly because of the unfavourable climatic conditions and prevalence of low-emitting PFTs (Table S1). China and the U.S. show large discrepancies between estimates using different LULC databases, e.g., emissions from China vary from 9.5 Tg in ISOPGFW to 23 Tg with ISOPMOD, in the range of reported values

from previous studies (Stavrakou et al., 2014; Li et al., 2013). The national emission estimates for China and Indonesia were significantly lower in the study of Stavrakou et al. (2014) also using MEGAN-MOHYCAN, 7 Tg yr$^{-1}$ and 8 Tg yr$^{-1}$ over 2005-2012. This difference is largely due to reduced basal emissions rates adopted in that study for tropical forests over Asia (Stavrakou et al., 2014), based on flux measurements in the rainforest of Borneo (Langford et al., 2010).

Meteorology induces positive trends in all countries, of the order of 0.5-1.5% yr$^{-1}$ in the CTRL run (Table 6) and is the main driver of the overall trends, except in Russia for which the positive trend induced by LULC changes according to MODIS (0.76% yr$^{-1}$) exceeds the meteorological effect (0.57% yr$^{-1}$). LULC changes lead to a reduction of the trends in the U.S., Brazil and Indonesia. This reduction is most significant for ISOPGFW over Brazil (-0.53% yr$^{-1}$) and Indonesia (-0.7% yr$^{-1}$). The emission trends over southern China are of opposite sign in ISOPMOD and ISOPGFW, consistent with the increasing TC

trend in MODIS and decreasing trend in GFWMOD shown on Fig. 3.

It is important to emphasize that the LULC-induced trend estimates given above are based on simulations assuming that the basal emission factor of each PFT remains constant. This assumption is not always verified, as e.g. the proportion of high isoprene emitters might change over time, leading to significant variations of the basal emission rate. Note also that the spatial

patterns of the emissions differ from those of inventories such as CAMS-GLOB-BIOv1.1 and the BIRA-IASB dataset since they used the gridded emission factor distributions instead of PFT-specific approach. In particular, the gridded distribution of the emission factor gives rise to two hotspots not present in the inventories presented here; one is located in the South Amazon Basin and the other in the Northern Australia (Sindelarova et al., 2014). The conversion of primary forests to tree plantations or secondary forests regrow may tend to increase the isoprene emission factor (Harley et al., 1999; Geron et al., 2006). As a

result, an increase in the isoprene emitting fraction of trees could easily compensate for the decrease in total number of trees. For instance, the rapid land use change and forest plantation establishment in South Asia, enhances the proportion of high isoprene-emitting species (Misztal et al., 2011).

### 4.3 Comparison to previous studies

At the global scale, only one study was conducted to assess isoprene emission responses to changes in TC by means of satellite

land cover datasets. Using the MEGANv2.1 with the Landsat tree cover continuous fields of Sexton et al. (2013) and fixed meteorological fields, Chen et al. (2018) estimated global isoprene emission at around 480 Tg yr$^{-1}$ in 2000. This estimate is higher than our estimate (350 Tg yr$^{-1}$ in 2001; cf. Table 5) based on Landsat TC data from Hansen et al. (2013). This disparity owes to differences in modelling settings and drivers. In particular, Chen et al. (2018) used the coupled MEGAN2.1-CLM4.5

along with satellite vegetation data instead of the modelled land cover from CLM4.5, the canopy environmental model from
Guenther et al. (2012) instead of the MOHYCAN model, and emission factors calculated using three vegetation categories,
namely the broad- and needle-leaved trees and non-trees instead of the PFT-dependent emission factors. The decreasing
emission trend induced by LULC evaluated for ISOPGFW amounted to 0.33% (Table 5), which is more than thrice the
decreasing trend of 0.1% $yr^{-1}$ over 2000-2015 in the study of Chen et al. (2018), with significant regional variations. The tree
cover trends in Chen et al. (2018) and GFWMOD are of opposite signs over West Africa (Gabon and Cameroon), India,
Yucatan Peninsula and southeastern U.S., whereas a qualitative agreement is found regarding the positive trends seen over
Europe and the Atlantic Forest in Brazil and the negative trends in Southeast Asia.

While the present study showed that meteorology is the main driver of emission trends at the global scale and in most forested
countries, a converse conclusion was drawn by Wang et al. (2020) who estimated that land cover changes caused a significant
emission trend over China during 2001-2016 (1.35% $yr^{-1}$), which dominates largely over the slight negative trend due to
meteorology. Wang et al. (2020) used the MODIS PFT dataset as in the present study, but with a different approach for the
conversion to MEGAN PFTs. Furthermore, their estimation accounts for changes in the soil moisture stress factor and in the
$CO_2$ inhibition effect. The annual isoprene emission for China amounted to 7.56 Tg of isoprene (average over 2001-2016), a
factor 3 lower than the 23 Tg $yr^{-1}$ estimated in ISOPMOD. In a previous study using both satellite retrieval and land use survey,
Fu and Liao (2014) estimated the emissions from China at 14.5 Tg $yr^{-1}$ in the mid-2000s and evaluated the LULC impact at
– 0.175% $yr^{-1}$ from the late 1980s to the mid-2000s, well below the trend due to meteorology (+0.85% $yr^{-1}$).

## 5 Evaluation with OMI HCHO observations

Three global simulations with the IMAGESv2 model are performed over 2005-2016. The period is selected so as to coincide
with HCHO data availability from the OMI satellite. The biogenic isoprene emissions used in those runs are described in the
previous section, except that the inhibition effect of $CO_2$ parameterized following Possell and Hewitt (2011) is now taken into
account. The three runs are: run A, using the CTRL emissions; run B, using the ISOPMOD emissions; and run C, using the
ISOPGFW emissions.

The interannual variability of seasonally-averaged modelled HCHO columns correlates very well with the OMI data, as shown
in Fig. 8 which displays the global distribution of the correlation coefficient of observed and calculated (run A) seasonally
averaged HCHO columns. In tropical regions, the averages are taken over June-July-August (JJA) for the northern Tropics (0-
30°N), and February-March-April (FMA) for the 0°-30°S band, corresponding to months with usually minimal biomass
burning activity (Fig. S6 and S7). At extra-tropical latitudes, the seasonal averages are calculated over the summer months,
when biogenic emissions and HCHO columns are the highest. High and statistically significant correlation coefficients are
found over most vegetated areas, where BVOC emissions are the dominant source of HCHO. This result essentially validates





the modelled response of BVOC emissions to short-term meteorological variability (Stavrakou et al., 2018). Whereas the impact of biomass burning on the correlation was mitigated in tropical areas due to the choice of the averaging period (Fig. S6 and S7), the high correlation of modelled and OMI columns in the boreal regions of Canada and Siberia is partly due to the impact of fire activity on HCHO (Fig. S7). Nevertheless, even at those latitudes, the interannual variability of biogenic

emissions contributes substantially to the correlation with OMI data (Stavrakou et al., 2018).

Certain forested regions exhibit positive but relatively low correlations, such as Ivory Coast, South China and New Guinea, for reasons unclear. Negative or very low coefficients are found in arid and semi-arid regions such as Northern Africa, Middle East, South Africa, Texas, Arizona, Northern Mexico, and Western Australia. Although this poor agreement might be partly

due to a relatively low HCHO signal and low biogenic emissions (e.g. over Sahara), the omission of a soil water stress factor in the parameterization of biogenic isoprene emissions likely contributes much to the discrepancy. The effect of LULC changes has little impact on correlations, with changes within the $\pm 0.1$ range (Fig. S8), indicating that interannual variability of isoprene and HCHO is dominated by meteorological variability, whereas LULCCs play only a minor role.

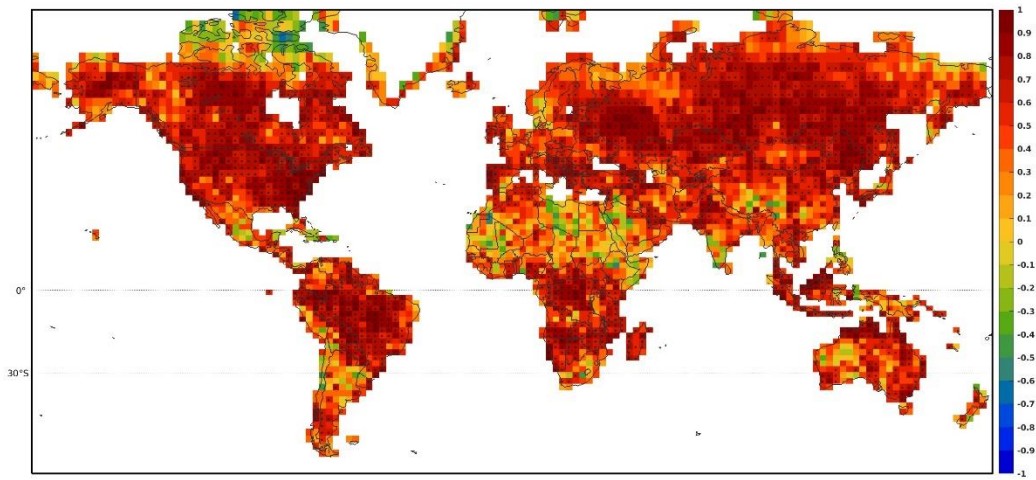


**Figure 8: Pearson's coefficient of correlation between the observed evolution of seasonally-averaged HCHO columns during 2005-2016 and the model-calculated values (run A). The seasonal averages are calculated over months June-August in the Northern Hemisphere (> 0° N), February-April in the 0°-30° S latitudinal band and December-February below the 30th parallel.  The stippling represents statistically significant correlation coefficient (p < 0.05).**

The interannual evolution of normalized seasonally-averaged datasets over five selected regions (Fig. S9) is shown in Fig. 9. Over Southeastern U.S. and South China, the relatively high HCHO columns of run B (Fig. S10) lead to greater fluctuations, e.g. in 2011 and 2013 over S-E U.S., that are not found in the observations, as seen from the higher RMSD for this simulation. Despite the inclusion of the inhibition effect due to $CO_2$, the simulations show stronger positive trends than the OMI observations in all five regions. Over Mato Grosso and South China, the positive trends in tree cover from the MODIS PFT

dataset enhance the positive HCHO trends and therefore worsen the agreement with OMI. On the other hand, the larger TC





negative trends of the GFWMOD-based simulation (run C) lead to a better agreement with trends from OMI data in all regions, especially in Mato Grosso, South China and Indonesia. In particular, the overestimation by 0.4% yr⁻¹ found in the control simulation over Indonesia and Mato Grosso is greatly improved, due to an important offset by 0.3 and 0.2% yr⁻¹, respectively. The overestimation of 1% yr⁻¹ in modelled HCHO trends over Equatorial Africa is largely due to abnormally low modelled
columns between 2006 and 2009, for reasons unclear. Nevertheless, interannual variability still shows a high correlation (> 0.8). The effect of LULC changes in this area does not bring a significant improvement on trends.





**Figure 9: Interannual variability of seasonally-averaged HCHO columns normalized by their 2005-2016 average over selected regions (Fig. S9). The seasonal averages are calculated over Jun-Aug in the Northern Hemisphere, and over the wet season (Feb-Apr) at Southern latitudes. Normalized OMI HCHO columns are represented by symbols with their 1-σ uncertainty bars; model results are shown as solid lines: run A (red), B (blue) and C (green). The correlation coefficients (R), root-mean square deviations (RMSD) multiplied by 100, and trends (in % yr$^{-1}$) with 1-σ uncertainty estimates are given inset.**



## 6 Summary and Conclusions

In this study, the tree cover distribution and trends from three global satellite-based LC products were intercompared for the
period 2001–2016 and put in perspective with national estimates from the FRA compilation. Two global isoprene emission inventories, ISOPMOD and ISOPGFW, were developed using the MEGAN-MOHYCAN model, with annual PFT distributions derived from satellite-based LC products for the 2001-2016 period. The impact of LULCC on emissions was estimated for both inventories. Finally, the interannual variability of isoprene emissions was evaluated through comparisons of modelled HCHO columns using the global CTM IMAGESv2 with spaceborne OMI HCHO observations over 2005-2016.
The main findings of this study are presented below.

- The global total TC area differs by more than 60% between the datasets, with 30.6 Mkm² (ESA), 32.2 Mkm² (GFWMOD) and 52.6 Mkm² (MODIS). Higher canopy densities and extent contributed to the large figure found with MODIS. Great disparities were also seen in global TC trends. Negative trends are found in the ESA dataset (-0.05% yr$^{-1}$) due to overall weak and sparse net changes, and in the GFWMOD dataset (-0.26% yr$^{-1}$) mostly due to net losses in high density forest canopies. A slightly net positive trend is derived from MODIS (0.03% yr$^{-1}$) due to the increase in areal coverage of temperate and boreal trees. Over the U.S., the reported positive trends in forest cover are contradicted by all remotely-sensed datasets. Over China and Russia as well, the positive trends from national estimates are at odds with the GFWMOD dataset, whereas an overall consistency between the national reports and the satellite-based estimates is found over Brazil and Indonesia.

- The global isoprene emissions estimated using the MEGAN-MOHYCAN model amount to 354, 418 and 520 Tg yr$^{-1}$ for ISOPGFW, CTRL and ISOPMOD averaged over 2001-2016, respectively. Strong dissimilarities are found in isoprene-rich areas (northeastern Brazil, South China, Eastern US, and Equatorial Africa) and stem from differences in tree coverage.

- The global impact of LULCCs is a mitigating effect on the strong positive trends (0.94% yr$^{-1}$) of isoprene emissions driven by meteorological parameters, primarily temperature and solar radiation. The cutbacks were estimated at 0.04% yr$^{-1}$ for ISOPMOD and 0.33% yr$^{-1}$ for ISOPGFW, due to the decreasing trends in tree coverage, in particular for broadleaf trees in tropical regions. Despite the slightly positive trend (+0.03% yr$^{-1}$) in global TC area in MODIS, decreases in tropical broadleaf tree cover led to a decline in global isoprene emissions, owing to the dominance of tropical forests to the global total. LULCCs have little impact on the interannual variability of isoprene emissions.

- The interannual variability of seasonally-averaged HCHO columns calculated by IMAGESv2 correlates very well with the OMI data. High and statistically significant correlation coefficients (R > 0.9) are found over most forested





areas, where BVOC emissions are the dominant source of HCHO. The model performance worsens over arid and semi-arid areas, likely due to the neglect of soil moisture stress effects in the MEGAN model setup used in this study. The effect of LULC changes has little impact on correlations, with changes within the ±0.1 range; the interannual variability is dominated by meteorological variability. The larger TC negative trends of the GFWMOD-based

simulation (run C) lead to a better agreement with trends from OMI data in all five regions, and in particular over Indonesia and Mato Grosso.

Although remote sensing is a tool of choice for constraining LULC in biogenic emission models, the present study casts light on the high uncertainties in the calculation of isoprene emissions associated with the representation of PFTs from satellite-

based LC products. The coarse resolution, the use of a discrete approach in the classification and the cross-walking process constitute notable limitations of satellite-based LULC maps underscored in this study. In the remote sensing community, the accurate identification of LULC maps is a widely researched topic and future developments of LULC maps could provide the modelling community with a more accurate mapping of PFTs that could contribute to improved estimation of isoprene emissions. Remotely-sensed PFTs would solve the problem of arbitrariness in the cross-walking into biome-based PFTs (Sun

and Jiang, 2008; Ustin and Gamon, 2010). But, to our knowledge, only few attempts were made to mitigate the issue (Oleson and Bonan, 2000; Bonan et al., 2002) and no such maps has been made publicly available at a global and long-term scale. Also, the current approach based on discrete classification could be replaced by continuous fields products, designed to overcome the spurious abrupt distinctions between land cover classes. Bonan et al. (2002) argued for the necessity of remotely-sensed, spatially continuous distributions of coexistent vegetation. Such maps would reduce model sensitivity to resolution

and solve the problem of the arbitrariness of biome-based PFTs. For now, the integrated use of the continuous tree cover fields available at very high resolution from the Global Forest Watch database and the discrete LULC maps from MODIS PFT seems to be a reasonable trade-off. These maps could be improved when the announced updated version of the GFW dataset will become available that solves the current inconsistencies in the mapped interannual loss or when tree cover maps including net changes could be made available so that the assumption of linear gains would not be needed. Despite those limitations, we

recommend the integrated use of continuous fine-resolution tree cover fields, as those provided by the GFW database, in the modelling of biogenic emissions and their trends.

**Data availability**

The ESA CCI-LC maps can be viewed online and downloaded  from https://maps.elie.ucl.ac.be (ESA-CCI-LC, 2017), and is also available at https://cds.climate.copernicus.eu. The MODIS Land Cover Type Product MCD12Q1 is available at

https://lpdaac.usgs.gov (Friedl et al., 2019). The GFW dataset can be viewed online at www.globalforestwatch.org and is available for download at http://earthenginepartners.appspot.com (Hansen et al., 2013). The FAOSTAT datasets is available at http://www.fao.org/faostat. The OMI HCHO column data are publicly accessible at https://h2co.aeronomie.be and

http://qa4ecv.eu (De Smedt et al. 2015, 2017, 2018). The ALBERI inventory, GFWMOD-based isoprene emissions, is
available at https://repository.aeronomie.be/ (Opacka, B., and Müller, J.-F.: MEGAN-MOHYCAN global isoprene emissions
accounting for space-based land cover changes, BIRA-IASB [ALBERI dataset], doi: 10.18758/71021062, 2021). The
MEGAN-MOHYCAN isoprene emission inventories for CLM and MODIS are publicly accessible at the BIRA-IASB
emission portal https://emissions.aeronomie.be.

**Supplement**.

The supplement to this article is available online at: **XXX.**

**Author contribution**

BO developed the PFT datasets, conducted the analysis, and drafted the paper. JFM and TS supervised the study and realized
the MEGAN-MOHYCAN and IMAGESv2 simulations. MB helped with data processing and graphical routines. AG, KS and
JM commented and provided feedback on the final results and the manuscript.

**Competing interests**

The authors declare that they have no conflict of interest.

**Acknowledgments**

This research was supported by the Belgian Science Policy Office (BELSPO) through the STEREO III programme (ALBERI
– Assessing Links between Biogenic Emissions and Remotely-sensed photosynthesis Indicators, contract no. SR/00/373
(2019-2021), and the PRODEX TROVA (2019-2020) project funded by the European Space Agency via the Belgian Science
Policy Office.

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
