# Peer review of "Global and regional impacts of land cover changes on isoprene emissions derived from spaceborne data and the MEGAN model"

_Atmospheric Chemistry and Physics, 2021_

## Author Comment (AC1)

**Reply to Referee#1**

**We would like to thank the reviewer for his/her positive evaluation of the manuscript and for the useful comments and suggestions. Below we address the raised concerns. The reviewer's comments are shown in bold and our replies are given in blue and additions to the original text in green.**

First, we mention the following correction: The third panel of Figure 3 (GFWMD trend) has been replaced as there was an error in the upload of the figure.

In addition, we have added Table S5 in the Supplement, which provides absolute and relative tree cover trends over 2001-2016 for a selection of 25 countries worldwide as derived from MODIS, ESA, and GFWMOD datasets.

**L. 103: "… at standard canopy conditions": Are these standard canopy conditions some sort of extension of the standard conditions (L. 97) used to define emission factors? From the equation, it appears that there should be standard LAI, phenology and age? Maybe this information can be added here.**

Yes, L.97 and L.103 refer to the same standard conditions. This is now clarified as follows.

L.100: "where the emission factor $\varepsilon$ ($\mu$g m$^{-2}$h$^{-1}$) represents the emission rate at standard conditions. The latter specify all relevant meteorological (temperature, solar radiation, air humidity, soil moisture, wind speed, …) and phenological (leaf area index and leaf age) variables, as defined by Guenther et al. (2006)."

**L. 122: "this step is a cause of uncertainty" – I agree, but this also applies to the original PFT distribution (L. 111-116). Are you able to comment on possible differences between the method used for MEGANv2.1 and the new classification? Is the "PFT scheme" in L. 164 the one in MEGANv2.1?**

It is true that any approach and each step bear uncertainties. The following change is applied in the text (L.122→ 126):

L.126: " The uncertainty of this step is mainly due to the relative arbitrariness of the cross-walking land cover legend tables resulting from the sometimes ambiguous definitions of the biome classes. "

The PFT scheme (L.164→ L.167) is the same for all LULC datasets. However, datasets differ in their definitions of the classes (as explained in Table 4 and Section 3.3). The method used for MEGANv2.1 is described in the NCAR Technical notes in Section 15.3.3 (Oleson et al., 2010). Basically, the PFTs (NET, NDT, BET, BDT, shrubs, grass, crops) were obtained by combining various products including MODIS VCF, MODIS Land Cover and Ramankutty et al. (2008). The method for deriving PFTs in our study was based on the cross-walking of the original biome-based classes *(LCCS classes for MODIS and ESA),* as described in our manuscript. The impact of the two approaches is difficult to assess since many factors are at play as discussed in the Section 3.3 of the manuscript (acquisition methods, mapping methodologies, original classification, cross-walking). Moreover, the definitions on which the CLM PFTs were based on are not available in the technical notes.

However, it is possible to evaluate the differences in the mapping methods used for the subdivision into climate zones and C3/C4. We conducted a test on the CLM map by aggregating classes with

[Figure]

*Figure: Arctic C3, Cool C3 and Warm C4 grass distributions (in %) as obtained using our method based on Poulter et al. (2011) (left panel) and difference between those distributions and the original distributions in CLM4 PFT (right panel) based on the mapping method of Still et al. (2013) using LAI instead of NDVI.*

common physiognomy, leaves and phenology (i.e., NET, NDT, BET, BDT, grass and shrub) and applying our method (described in Section S3). The resulting sub-classes were compared to the original sub-classes, as shown in the figure above. Overall, except for the grass sub-classes, little differences were seen. Our method for climate subdivisions based on Poulter et al. (2011) and the method of Nemani and Running (1996) gave very similar results. But these changes have negligible impact on global isoprene emissions given the very low basal emission factors of grass PFTs.

A comment has been added in the supplementary material (Section S3) and mentioned in section 2.1.

**L. 324: The discussion about tree definitions is valuable to understand (part of) the differences between the datasets. How does this affect the application to simulation of isoprene later on in the manuscript? What is the difference in the emissions if plants e.g. are defined as tree or shrub? L. 369 attributes an important role to tree cover, but the difference in the emissions seems to depend very much on the type of tree/shrub (Table S1). I welcome the authors to extend the discussion towards the application of these data sets to isoprene emissions.**

We agree with the reviewer that the tree cover distribution is not sufficient and must be complemented by an additional PFT dataset for further discrimination between tree/shrub/grass types (here the MODIS PFT distribution). Note however that besides needleleaf trees, all tree PFTs emit much more strongly than non-tree PFTs (Table S1). Since broadleaf trees are the dominant tree PFTs in tropical and temperate zones, where temperature and PAR conditions are most favourable to the emissions,

the tree cover fraction is clearly the most important factor determining the emissions (besides meteorology). In boreal regions, we agree that tree cover plays a less important role and that the discrimination between different tree PFTs and between non-tree PFTs becomes crucial.

**L. 343: "fire-over areas" – do you mean burnt areas?**

The sentence has been changed to (L.343 → 344):

L. 344: "Nursery land and  clear cut or burnt areas that do not meet the biophysical requirements stated by the FAO user guide are included as well as economic and bamboo forests. "

**Fig. 5: The emission maps for the three simulations are very similar. I would prefer to see the results from CTRL, combined with differences between ISOPMOD-CTRL and ISOPGFW-CTRL (as in the right-hand panels of Fig. S3). This would help to understand the differences between the three simulations.**

We thank the reviewer for the useful advice. The figures were changed accordingly.

**L. 457: What is meant with "weakly dependent on variations in PFTs"? Does it mean that the variability is modulated somehow by the land cover sets used, hence leading to differences between simulations, or is there a variability within one of the simulations because of IAV of the cover fractions of the PFTs?**

It is meant that the variability is (weakly) modulated by the choice of land cover dataset, hence leading to small differences between simulations. It is now clarified in the text (L.457→ 460):

L. 460: "This variability is only weakly dependent on the choice of land cover dataset."

**Table 6: I would suggest to remove "mitigating" from the last line of the table header, because LULC does not always reduce emissions.**

Corrected.

**L. 540: Why is the isoprene inhibition accounted for in the CTM simulations, but not in the results presented earlier? Given that the changes in CO2 are limited for the period used in the simulation, I do not expect it to have a large effect on the outcome, but it brings an inconsistency into the study.**

The $CO_2$ inhibition effect is very uncertain and for this reason was neglected in the core of this study (Sect. 3 and 4). It was accounted for in the HCHO simulations as it has a substantial offsetting effect on isoprene trends (-0.5 %yr$^{-1}$) which improves the agreement with the HCHO trends.

The impact of $CO_2$ inhibition (and soil moisture stress) is briefly discussed in Sect. S6 in the Supplementary material and mentioned in line L.452 of Sect. 4.

The following clarification and correction are added in Section 4:

L. 452: "Note that the $CO_2$ inhibition effect, not considered in those simulations, would further offset global trends by about  0.5% yr$^{-1}$ according to the parameterization of Possel and Hewitt (2011), whereas the soil moisture stress has little impact on trends (Table S6 and Sect. S6)."

We added the following clarification at the beginning of Section 5:

L.542: "Three global simulations with the IMAGESv2 model are performed over 2005-2016. The period is selected so as to coincide with HCHO data availability from the OMI satellite. The biogenic isoprene emissions used in those runs are described in the previous section, except that the inhibition effect of $CO_2$ parameterized following Possell and Hewitt (2011) is now taken into account. Although very uncertain, its inclusion is motivated by its substantial effect on isoprene trends (discussed and quantified in Sect. S6), which improves the agreement with OMI HCHO trends. The three runs are: run A, using the CTRL emissions; run B, using the ISOPMOD emissions; and run C, using the ISOPGFW emissions."

**Fig. 9: Maybe you could repeat the different land cover products in the figure caption (together with the A, B and C simulations).**

Done.

**L. 619: The three CTM simulations show very similar results (Fig. 9), and I agree that it is meteorological drivers rather than LULC changes that are responsible for this. I would suggest to extend the statement that the GFWMOD-based simulation results in a better agreement (L. 619) - while formally correct - with a remark on this small difference.**

Agreed. The text is modified as follows (L.619 → L.624):

L.624: "Although the three model simulations show similar results, the larger TC negative trends of the GFWMOD-based simulation (run C) lead to a better agreement with trends from OMI data in all five regions, and in particular over Indonesia and Mato Grosso."

**The print quality of some of the maps could be increased by using a higher resolution or by using vector graphics, in particular Fig. 3, and for the stippling in Fig. 8.**

Figures with increased resolution will be provided for the final (online) version of the manuscript.

---

## Author Comment (AC2)

**Reply to Referee#2**

**We would like to thank the reviewer for his/her positive evaluation of the manuscript and for the useful comments and suggestions. Below we address the raised concerns. The reviewer's comments are shown in bold and our replies are given in blue and additions to the original text in green.**

First, we mention the following correction: The third panel of Figure 3 (GFWMD trend) has been replaced as there was an error in the upload of the figure.

In addition, we have added Table S5 in the Supplement, which provides absolute and relative tree cover trends over 2001-2016 for a selection of 25 countries worldwide as derived from MODIS, ESA, and GFWMOD datasets.

**Line 20: "At national level, the increasing trends in forest cover reported by some national inventories (in particular for the US) are contradicted by all remotely-sensed datasets". I anticipate this section will peak substantial interest for a variety of audiences. I suggest adding a brief clause as to the cause of the discrepancies (a short reference to section 3.3).**

The following is added in the abstract of the revised manuscript:

L. 20: "At national level, the increasing trends in forest cover reported by some national inventories (in particular for the US) are contradicted by all remotely-sensed datasets. To a great extent, these discrepancies stem from the plurality of definitions of forest used. According to some local census, clear cut areas, seedling or young trees are classified as forest while satellite-based mappings of trees rely on a minimum height."

**Figure 9 and supporting discussion: Zhu et al. (2017) attribute trends in HCHO in the Northwestern US to increasing forest cover. Is this compatible with your results?**

In Zhu et al. (2017), an increasing HCHO trend (5.4% $yr^{-1}$) was derived over a few 0.5°×0.5° pixels in the Northwestern US, located within a large box (~39°-45°N and 124°-120°W). The trend was attributed to a large increasing trend (4.3% per year) of the needleleaf evergreen trees (NET) over 2005-2014 estimated using MODIS land cover data.

In our analysis, MODIS NET shows only a small increasing trend of 0.17% $yr^{-1}$ when considering the aforementioned box, and a decreasing trend is found based on the 4 points (-0.09% $yr^{-1}$). The GFWMOD shows strong decreasing trends of from -0.3 to -0.4 % per year in either case. Those results are therefore very different from those given by Zhu et al. (2017), for reasons unclear but possibly related to the earlier version of MODIS land cover data used in Zhu et al (2017).

Furthermore, the QA4ECV OMI HCHO averaged over May-September showed lower positive trends over this area based on either the 4 cells (2.87% $yr^{-1}$) or the entire box (1.13% $yr^{-1}$). Note however that we did not apply any temperature-based correction to HCHO columns as in the study of Zhu et al. (2017).

**Can the authors briefly comment on discrepancies in the magnitude of monoterpene emission trends (as in the summary, second bullet, or as in Figure 7)? While not the focus of this work, the results would be interesting given the high variability in trends in the northern latitudes.**

Isoprene is the focus of this work. Monoterpenes are included in the IMAGES model calculations, based on MEGAN estimates using gridded basal emission rates. The latter is not adequate for the analysis of the impact of LULC changes on monoterpene emission trends.

**In section 5, $CO_2$ inhibition is turned on, whereas it is neglected previously. This leads to some confusion as to whether the emission trends presented previously apply to the HCHO trends shown here. I suggest incorporating the $CO_2$ inhibition factor throughout.**

The $CO_2$ inhibition effect is very uncertain and for this reason was neglected in the core of this study (Sect. 3 and 4). It was accounted for in the HCHO simulations as it has a substantial offsetting effect on isoprene trends (-0.5 %yr$^{-1}$) which improves the agreement with the HCHO trends. The impact of $CO_2$ inhibition (and soil moisture stress) is briefly discussed in Sect. S6 in the Supplementary material and mentioned in line L.452 of Sect. 4. The following clarification and correction are added in Section 4 of the revised manuscript:

L. 452: "Note that the $CO_2$ inhibition effect, not considered in those simulations, would further offset global trends by about  0.5% yr$^{-1}$ according to the parameterization of Possell and Hewitt (2011), whereas the soil moisture stress has little impact on trends (Table S6 and Sect. S6)."

We added the following clarification at the beginning of Section 5:

L. 542: "Three global simulations with the IMAGESv2 model are performed over 2005-2016. The period is selected so as to coincide with HCHO data availability from the OMI satellite. The biogenic isoprene emissions used in those runs are described in the previous section, except that the inhibition effect of $CO_2$ parameterized following Possell and Hewitt (2011) is now taken into account. Although very uncertain, its inclusion is motivated by its substantial effect on isoprene trends (discussed and quantified in Sect. S6), which improves the agreement with OMI HCHO trends. The three runs are: run A, using the CTRL emissions; run B, using the ISOPMOD emissions; and run C, using the ISOPGFW emissions."